# Natural Radioactivity of Laterite and Volcanic Rock Sample for Radioactive Mineral Exploration in Mamuju, Indonesia

**Ilsa Rosianna [1,2,†], Eka Djatnika Nugraha [3,4,†], Heri Syaeful [1], Sugili Putra [2], Masahiro Hosoda [4,5], Naofumi Akata [5] and Shinji Tokonami [5,*]**

1 Centre for Nuclear Minerals Technology, National Nuclear Energy Agency of Indonesia (BATAN), Jakarta 12440, Indonesia; ilsa.r@batan.go.id (I.R.); syaeful@batan.go.id (H.S.)
2 Polytechnic Institute of Nuclear Technology (STTN), Yogyakarta 55281, Indonesia; sugiliputra@gmail.com
3 Centre for Technology of Radiation Safety and Metrology, National Nuclear Energy Agency of Indonesia (BATAN), Jakarta 12440, Indonesia; eka.dj.n@batan.go.id
4 Department of Radiation Science, Graduate School of Health Sciences, Hirosaki University, Hirosaki, Aomori 036-8560, Japan; m_hosoda@hirosaki-u.ac.jp
5 Institute of Radiation Emergency Medicine, Hirosaki University, Hirosaki, Aomori 036-8560, Japan; akata@hirosaki-u.ac.jp
* Correspondence: tokonami@hirosaki-u.ac.jp; Tel.: +81-172-39-5404
† These authors have made an equivalent contribution to the first author.

**Abstract:** Mamuju is a region of Indonesia with relatively high exposure to natural radiation. Since 2012, Mamuju has been a uranium and thorium exploration area. Several mapping studies of the region have been carried out to depict NORM (naturally occurring radioactive material) areas and areas with uranium anomalies. This paper is the first study to use radioactivity measurements of laterite and volcanic rocks to determine the characteristics of radionuclides and other mineral measurements, which are essential for categorising Mamuju rocks and exploring the region as a potential mining area. The radioactivity of the samples was measured using a high-purity germanium (HPGe) detector. Furthermore, we used X-ray fluorescence (XRF) to determine the rock mineral composition. Mamuju is anomalous due to its high content of $^{238}$U and $^{232}$Th concentrations of 539–128,699 Bq·kg$^{-1}$ (average: 22,882 Bq·kg$^{-1}$) and 471–288,639 Bq·kg$^{-1}$ (average: 33,549 Bq·kg$^{-1}$), respectively. The major elements are dominant, including $Fe_2O_3$, $SiO_2$, $Al_2O_3$, and $Na_2O$, with several other major elements such as $MnO$, $TiO_2$, and $CaO$.

**Keywords:** uranium; radioactive mineral; exploration; rocks; radioactivity

## 1. Introduction

The government of Indonesia plans to build a nuclear power plant (NPP). The amount of nuclear fuel necessary for the plant's operation will require a supply of uranium ore. To maintain the continuity of the NPP's operation, it is essential to maintain the balance of the demand and supply of uranium. Therefore, before the NPP is built, it is necessary to analyse the availability of uranium to develop an adequate and sustainable uranium supply strategy.

Mamuju is a region of Indonesia with relatively high exposure to natural radiation. According to the results of research conducted by the Centre for Safety and Metrology Technology of Radiation—BATAN, the ambient dose equivalent rate in the Mamuju area reached 2800 nSv·h$^{-1}$ with an average value of 631 ± 23 nSv·h$^{-1}$. This value is quite high compared to the average natural radiation dose rate in Indonesia, which is approximately 50 nSv·h$^{-1}$ [1,2]. Furthermore, a mapping research was conducted

by the Centre for Nuclear Minerals Technology—BATAN to depict the NORM region and areas with uranium (U) anomalies in Mamuju [3]. Significant thorium (Th) levels were found in several areas in Tapalang, and high uranium (U) levels were observed at several sites.

U and Th contents in volcanic rocks in the Tapalang area are controlled by the distribution of rock types and their constituent soils. The main minerals that contain U and Th are davidite and thorianite, while gummite and autunite are secondary minerals [4]. Mamuju contains the Adang volcano rock complex, which is the product of the volcanism process of a volcano complex with an eruption centre and several lava domes [5]. The formation of volcanic rock complexes is related to the presence of radioactive minerals in basaltic–andesitic rocks. The accumulation of U and Th also occurs in volcanic rocks that have undergone weathering and enrichment processes [6].

Natural enrichment of radioactive minerals occurs due to volcanic activity in Mamuju area. Remote sensing analysis shows that the Mamuju area, including Tapalang, is a centre of volcanic activity, revealing a number of circular features throughout the region [7]. Areas that have high Th levels generally have a high level of rock alteration with a varied Th/U ratio.

U deposits can be formed under various geological conditions, including high-grade metamorphic, plutonic, metasomatic, hydrothermal, and basin diagenesis, and in volcanic, sedimentary, and superficial environments [8]. According to these geological processes, U ore deposits are divided into dozens of groups on the basis of different criteria. In intrusive-type U deposits, the mineralisation is associated with high-temperature magmatic systems such as alaskite, pegmatite, and granite [9]. The main U mineral in this group is typically magmatic uraninite (or uranothorite in some cases) [10]. The ore grades of this type of U deposits are usually low (0.01–0.1%). Mamuju and the surrounding geological area are composed of the Latimojong formation, Talaya volcanic rocks, Adang volcanic rocks, intrusion rocks, the Mamuju formation, the Tapalang member of Mamuju formation, and alluvium deposits (Figure 1) [4].

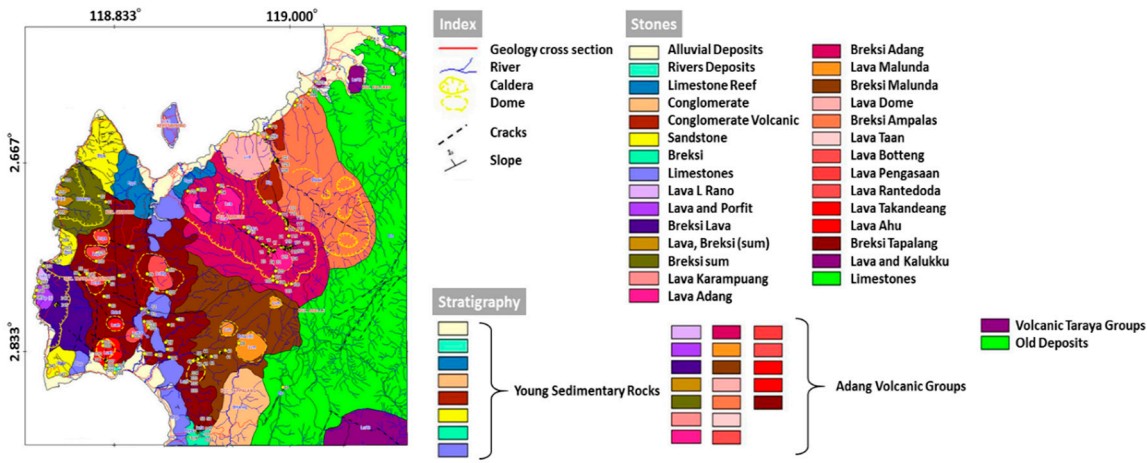

**Figure 1.** The geological map of the Mamuju area.

U has several naturally occurring isotopes, including $^{238}$U, $^{235}$U, and $^{234}$U. The determination of the $^{238}$U/$^{235}$U activity ratio (AR) has been used as an indication of the presence of depleted uranium (DU) in the environment [11–14]; naturally occurring U has a constant $^{238}$U/$^{235}$U AR of 21.7 [15], while DU has a ratio above 76.9 [11]. DU is characterised as technologically enhanced naturally occurring. Natural U contains 0.7% of the $^{235}$U isotope. The remaining 99.3% is mostly the $^{238}$U isotope, which does not contribute directly to the fission process. Most reactors are light-water reactors (of two types, pressurised water reactor and boiling water reactor ) and require U to be enriched from 0.7% to 3–5% $^{235}$U in their fuel.

The objective of this paper is to describe the radioactivity characteristics of laterite and volcanic rock samples from Mamuju, which are essential to the categorisation of Mamuju rocks in the radioactive mineral exploration area. This is the first study of the radioactivity characteristics of laterite and

volcanic rock samples from Mamuju. Other researchers explained the geological order of the region [3], mapping radionuclides on the basis of satellite imagery results [7] and X-ray fluorescence (XRF) measurements [3–5]

## 2. Materials and Methods

### 2.1. Study Area

Mamuju is located on the western edge of the island Sulawesi. The topography of the Mamuju area includes the area from the coast to the mountain. The city of Mamuju lies in the province of West Sulawesi. Sulawesi Island has a complex geological setting with the complex of tectonic setting caused by the three interaction plates (Pacific, Eurasian continental, and Australian continental plates). Sulawesi island is generally divided into nonvolcanic arch and volcanic arch [16].

A total of 30 fresh rock samples were obtained from several subdistricts of Mamuju area: Takandeang, Rante Dunia, Kasambang, Botteng, Ahu, Hulu Mamuju, Hulu Ampalas, and Salumati (Figure 2). The purpose of taking fresh rock is to avoid weathering factors so that the fresh rock obtained is in accordance with the actual rock conditions. Fresh rock samples came from locations with high exposure; then, outcrop observations were made using a magnifying glass and ultraviolet flashlight. After the presence of mineral deposits in the outcrop was identified, further sampling was carried out at that location, accompanied by recording the coordinates of the location, the depth of rock or soil, and the rock and soil types.

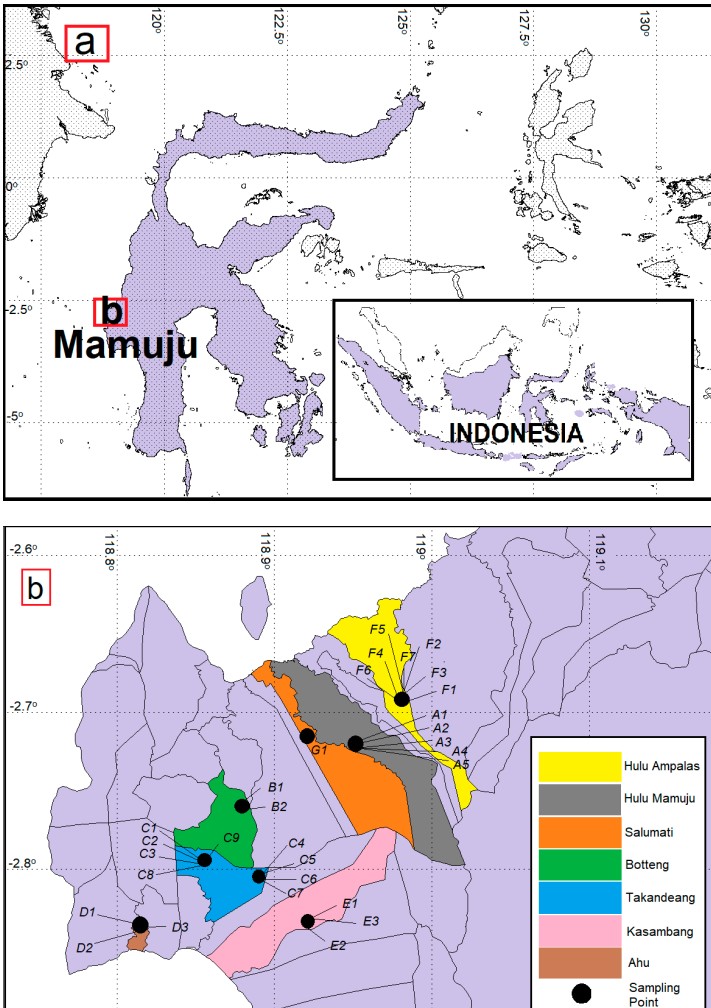

**Figure 2.** Study area: (**a**) Sulawesi Island and Mamuju area; (**b**) Mamuju subdistrict and sampling point.

## 2.2. Identification of Elements

We used XRF (Xepos, Ametek, Berwyn, PA, USA) to determine the elements in laterite and volcanic rock samples. The samples were air-dried at 25–35 °C for three days and then dried in an oven at 105 °C for more than 24 h to remove the moisture content until a constant weight was reached. Then, the samples were crushed and mashed to a size of 200 mesh and heated with 105 °C for 2 h to remove organic compounds. A total of 5 g of each sample was weighed, and 1 g of lithium tetraborate ($Li_2B_4O_7$; Merck, Darmstadt, Germany) was added as a binding agent. The mixture was stirred well and compressed into pellet form with a hydraulic press. Once the samples were shaped, the pellets were analysed using XRF. Before testing the samples, we performed XRF calibration using a blank, OREAS-124 (Mantra Resousces Nyota Prospect, Lindi, Tanzania), OREAS-465 (Tanzania- Mantra Resousces Nyota Prospect), CGL-124 (Central Geological Laboratory, Ulan Bator, Mongolia, RG-Th (IAEA), and GSR-5 (China).

## 2.3. Radioactivity Measurement

We measured the activity concentrations of the samples using an HPGe detector (GEM, ORTEC, USA). To prepare such a standard, a known quantity of a multi-nuclide standard source supplied by the Japan Radioisotope Association was placed in a cylindrical container of 48 mm × 55 mm (U-8 container). Approximately 100 g of each sample was weighed and sealed in U-8 container. The air-tightness of this vial aims to create equilibrium between the parent and the progeny, of which one of the decays of U and Th is $^{222}$Rn and $^{220}$Rn, which are gaseous. Furthermore, the samples are left for more than 30 days to allow decay products to reach secular equilibrium before further gamma spectrometer measurements. It was necessary to calibrate the counting system with a standard source with the same geometry as those of the samples. We measured $^{238}$U, $^{235}$U, $^{232}$Th, and $^{40}$K using a HPGe detector with a relative efficiency of 40%, a resolution of 1.81 keV at 1.33 MeV, and ultra-low background shielding using old lead content. The detector was enclosed in a 10 cm thick compact lead shield. The counts of the samples were obtained by analysing the spectra obtained from the Multi-Channel Analyzer (MCA) on a PC with associated Gamma Studio software (Seiko, EG&G, Tokyo, Japan). The counting time was about 1800–70,000 s, depending on when the total count of each radionuclide reached 10,000 to obtain adequate certainty. The full energy absorbed peaks of 351 for $^{214}$Pb and 609 keV for $^{214}$Bi were identified for calculations of $^{238}$U or $^{226}$Ra concentration. $^{235}$U radioactivity was determined using the gamma energy lines of 143 keV and 163 keV. However, the most intensive gamma line, 185 keV, was excluded to avoid interference with the 186 keV gamma energy from $^{226}$Ra [17]. The full energy absorbed peaks of 238 keV for $^{212}$Pb, 581 keV for $^{208}$Tl, and 911 keV for $^{228}$Ac were used for $^{232}$Th, and a single peak of 1460 keV was used for $^{40}$K. The minimum detectable concentrations (MDCs) of $^{238}$U, $^{232}$Th, $^{235}$U, and $^{40}$K for this measurement were $5.2 \times 10^{-3}$, $5.2 \times 10^{-3}$, $3.4 \times 10^{-3}$, and $1.5 \times 10^{-3}$ Bq·kg$^{-1}$, respectively. We used Equations (1) and (2) to calculate activity concentrations from these measurements.

$$A = \frac{n}{E \, Y \, W f_c}, \tag{1}$$

$$L_D = L_C + K\sigma_D, \tag{2}$$

where $n$ is net count per second, $E$ is the counting efficiency, $Y$ is the energy yield, $W$ is the sample weight (kg), and $f_c$ is the correction factor (including summing in, summing out, decay factor, recovery factor, attenuation factor, branching ratio, and growth factor). $L_D$ is the detection limit, $L_C$ is the critical level below which no signal can be detected, $\sigma_D$ is the standard deviation, and $K$ is the error probability [18,19].

## 2.4. Statistical Analysis

We used Origin Pro 2020b (student version) to evaluate significant relationships between the distribution coefficients of radionuclides and major elements and their oxide contents. We conducted

a Pearson correlation analysis and calculated the values of Pearson correlation coefficients with a two-tailed significance test (*p*-value at 0.05 and 0.01). Equation (3) expresses the Pearson correlation coefficients [20].

$$r = \frac{\sum_{i=1}^{n}(x_i - \bar{x})(y_i - \bar{y})}{\sqrt{\sum_{i=1}^{n}(x_i - \bar{x})^2} \sqrt{\sum_{i=1}^{n}(y_i - \bar{y})^2}}$$

(3)

where *n* is the sample size, $x_i$ and $y_i$ are the individual sample points indexed, and $\bar{x}$, $\bar{y}$ are the sample means.

## 3. Results

According to the regional geological data, the volcanic rocks found in the study area belong to the Talaya volcano group and the Adang volcano group. In general, sedimentary rocks in Mamuju occur in laterite, strata, and structures, while the constituent rocks of the volcanic complex in Mamuju Region are varied; thus, it is necessary to use very fresh rock samples. The base rock is generally dominated by types of tephrite, tephriponolite, phonotephrite, and phonolite (Table 1).

**Table 1.** Details of rock sample.

| Sample No. | Samples Code | Sedimentation Type | Rock Type | Rocks Name | GPS (UTM) | | | Location |
|---|---|---|---|---|---|---|---|---|
| | | | | | Zone | East (m) | North (m) | |
| 1 | A1 | Structure | Lava phonolite | Lava phonolite | 50 S | 721,567.00 | 9,696,880.00 | Hulu Mamuju |
| 2 | A2 | Structure | Lava phonolite | Lava phonolite | 50 S | 721,530.00 | 9,696,892.00 | Hulu Mamuju |
| 3 | A3 | Structure | Lava phonolite | Lava phonolite | 50 S | 721,774.00 | 9,696,416.00 | Hulu Mamuju |
| 4 | A4 | Structure | Lava phonolite | Lava phonolite | 50 S | 721,575.00 | 9,696,825.00 | Hulu Mamuju |
| 5 | A5 | Structure | Lava phonolite | Lava phonolite | 50 S | 721,827.00 | 9,696,745.00 | Hulu Mamuju |
| 6 | B1 | Strata | Phonolitoid | Phonolitoid | 50 S | 707,790.00 | 9,692,800.00 | Botteng |
| 7 | B2 | Strata | Phonolitoid | Phonolite | 50 S | 707,412.02 | 9,694,718.92 | Botteng |
| 8 | C1 | Structure | Autobreksi | Phonolite | 50 S | 705,746.00 | 9,688,196.00 | Takandeang |
| 9 | C2 | Structure | Autobreksi | Phonolite | 50 S | 705,746.00 | 9,688,196.00 | Takandeang |
| 10 | C3 | Structure | Autobreksi | Phonolite | 50 S | 705,746.00 | 9,688,196.00 | Takandeang |
| 11 | C4 | Laterite | Phonolitoid | Phonolitoid | 50 S | 709,486.00 | 9,689,203.00 | Rante Dunia |
| 12 | C5 | Laterite | Phonolitoid | Phonolitoid | 50 S | 709,486.00 | 9,689,203.00 | Rante Dunia |
| 13 | C6 | Laterite | Phonolitoid | Phonolitoid | 50 S | 709,486.00 | 9,689,203.00 | Rante Dunia |
| 14 | C7 | Laterite | Phonolitoid | Phonolitoid | 50 S | 709,486.00 | 9,689,203.00 | Rante Dunia |
| 15 | C8 | Structure | Autobreksi | Phonolite | 50 S | 705,746.00 | 9,688,196.00 | Takandeang |
| 16 | D1 | Laterite | Igneous | Phonolitee | 50 S | 703,255.00 | 9,689,161.00 | Ahu |
| 17 | C9 | Laterite | Igneous | Phonolite | 50 S | 704,674.00 | 9,689,566.00 | Takandeang |
| 18 | E1 | Strata | Sediment | Breksi | 50 S | 711,782.00 | 9,680,503.00 | Kasambang |
| 19 | E2 | Strata | Igneous | Phonolitoid | 50 S | 702,282.00 | 9,680,381.00 | kasambang |
| 20 | E3 | Strata | Igneous | Phonolitoid | 50 S | 712,050.00 | 9,680,507.00 | kasambang |
| 21 | D2 | Structure | Igneous | Phonolitoid | 50 S | 703,401.28 | 9,686,395.53 | Ahu |
| 22 | D3 | Strata | Igneous | Phonolitoid | 50 S | 703,734.50 | 9,685,494.98 | Ahu |
| 23 | F1 | Strata | Igneous | Phonolitoid | 50 S | 722,408.00 | 9,702,740.00 | Hulu Ampalas |
| 24 | F2 | Strata | Igneous | Phonolitoid | 50 S | 722,336.00 | 9,702,587.00 | Hulu Ampalas |
| 25 | F3 | Strata | Igneous | Phonolitoid | 50 S | 722,343.00 | 9,702,489.00 | Hulu Ampalas |
| 26 | F4 | Strata | Igneous | Phonolitoid | 50 S | 722,356.00 | 9,702,673.00 | Hulu Ampalas |
| 27 | F5 | Strata | Igneous | Phonolitoid | 50 S | 722,493.00 | 9,703,606.00 | Hulu Ampalas |
| 28 | F6 | Strata | Igneous | Phonolitoid | 50 S | 722,295.00 | 9,702,362.00 | Hulu Ampalas |
| 29 | F7 | Strata | Igneous | Phonolitoid | 50 S | 722,396.00 | 9,704,904.00 | Hulu Ampalas |
| 30 | G1 | Strata | Igneous | Phonolitoid | 50 S | 705,443.00 | 9,689,232.00 | Salumati |

Volcanic rocks from the Adang complex are composed of trachyte, tephra-phonolite, phono-tephrite, and phonolite rock with ultrapotassic affinity formed in the active continental margin (ACM) crust within the SW continental crust of Sulawesi [4,6]. The XRF analysis showed that

the rocks in Mamuju are dominated by the elements $Fe_2O_3$, $SiO_2$, $Al_2O_3$, and $Na_2O$, with several other major elements such as $MnO$, $TiO_2$, and $CaO$ (Table 2).

**Table 2.** Major elements.

| Sample | $Na_2O$ % | $MgO$ % | $Al_2O_3$ % | $SiO_2$ % | $P_2O_5$ % | $K_2O$ % | $CaO$ % | $TiO_2$ % | $Fe_2O_3$ % |
|---|---|---|---|---|---|---|---|---|---|
| 1 | 7.5 ± 0.2 | 0.3 ± 0.1 | 2.8 ± 0.1 | 11.6 ± 0.4 | 1.6 ± 0.1 | 2.8 ± 0.2 | 2.9 ± 0.2 | 3.2 ± 0.1 | 13.6 ± 0.6 |
| 2 | 6.5 ± 0.2 | 0.4 ± 0.1 | 16.4 ± 0.5 | 44.9 ± 1.4 | 1.1 ± 0.1 | 13.5 ± 0.5 | 0.5 ± 0.1 | 1.6 ± 0.2 | 10.1 ± 0.6 |
| 3 | 2.2 ± 0.1 | 0.4 ± 0.1 | 2.9 ± 0.1 | 10.9 ± 0.3 | 1.0 ± 0.1 | 3.3 ± 0.2 | 2.4 ± 0.2 | 0.6 ± 0.1 | 6.3 ± 0.4 |
| 4 | 2.0 ± 0.1 | 0.3 ± 0.1 | 2.9 ± 0.1 | 10.8 ± 0.3 | 0.8 ± 0.1 | 3.4 ± 0.2 | 2.8 ± 0.2 | 1.4 ± 0.1 | 14.3 ± 0.8 |
| 5 | 2.4 ± 0.1 | 0.4 ± 0.1 | 4.2 ± 0.1 | 14.2 ± 0.4 | 0.9 ± 0.1 | 5.1 ± 0.4 | 0.2 ± 0.1 | 0.2 ± 0.1 | 4.4 ± 0.3 |
| 6 | 4.3 ± 0.1 | 1.1 ± 0.1 | 7.2 ± 0.2 | 27.0 ± 0.8 | 1.1 ± 0.1 | 0.5 ± 0.1 | 4.5 ± 0.3 | 2.1 ± 0.1 | 11.7 ± 0.6 |
| 7 | 3.5 ± 0.1 | 0.3 ± 0.1 | 8.7 ± 0.2 | 10.9 ± 0.3 | 0.9 ± 0.1 | 0.7 ± 0.1 | 0.5 ± 0.1 | 3.6 ± 0.3 | 25.0 ± 1.8 |
| 8 | 3.8 ± 0.1 | 3.6 ± 0.1 | 14.8 ± 0.5 | 61.1 ± 1.8 | 0.2 ± 0.1 | 0.3 ± 0.1 | 2.1 ± 0.2 | 2.0 ± 0.2 | 10.9 ± 0.6 |
| 9 | 3.2 ± 0.1 | 4.5 ± 0.2 | 12.8 ± 0.5 | 50.7 ± 1.6 | 0.2 ± 0.1 | 0.2 ± 0.1 | 1.6 ± 0.2 | 1.6 ± 0.2 | 9.8 ± 0.6 |
| 10 | 3.6 ± 0.1 | 4.3 ± 0.2 | 14.4 ± 0.5 | 56.6 ± 1.7 | 0.2 ± 0.1 | 0.3 ± 0.1 | 2.0 ± 0.2 | 1.9 ± 0.2 | 9.9 ± 0.6 |
| 11 | 0.6 ± 0.1 | 0.5 ± 0.1 | 23.6 ± 0.7 | 39.5 ± 1.2 | 0.2 ± 0.1 | 0.9 ± 0.1 | 0.2 ± 0.1 | 2.9 ± 0.2 | 19.5 ± 1.1 |
| 12 | 0.4 ± 0.1 | 0.5 ± 0.1 | 22.4 ± 0.7 | 38.1 ± 1.2 | 0.2 ± 0.1 | 1.0 ± 0.1 | 0.2 ± 0.1 | 2.8 ± 0.2 | 18.7 ± 1.0 |
| 13 | 1.2 ± 0.1 | 0.9 ± 0.1 | 29.6 ± 0.8 | 47.9 ± 1.4 | 0.3 ± 0.1 | 0.9 ± 0.1 | 0.2 ± 0.1 | 2.7 ± 0.2 | 17.1 ± 1.0 |
| 14 | 0.8 ± 0.1 | 0.7 ± 0.1 | 29.4 ± 0.8 | 46.7 ± 1.4 | 0.2 ± 0.1 | 0.9 ± 0.1 | 0.2 ± 0.1 | 2.8 ± 0.2 | 19.0 ± 1.0 |
| 15 | 3.1 ± 0.1 | 5.5 ± 0.2 | 15.2 ± 0.5 | 58.6 ± 1.7 | 0.2 ± 0.1 | 0.2 ± 0.1 | 1.9 ± 0.2 | 2.0 ± 0.2 | 11.9 ± 0.6 |
| 16 | 5.9 ± 0.2 | 1.3 ± 0.1 | 15.4 ± 0.5 | 51.3 ± 1.6 | 1.0 ± 0.1 | 10.0 ± 0.5 | 5.4 ± 0.4 | 1.0 ± 0.1 | 8.8 ± 0.6 |
| 17 | 7.6 ± 0.2 | 2.4 ± 0.1 | 11.8 ± 0.4 | 48.8 ± 1.6 | 1.7 ± 0.1 | 3.0 ± 0.2 | 5.5 ± 0.4 | 1.9 ± 0.2 | 17.5 ± 1.0 |
| 18 | 3.4 ± 0.1 | 0.5 ± 0.1 | 6.4 ± 0.2 | 28.1 ± 1.0 | 0.7 ± 0.1 | 5.2 ± 0.3 | 2.4 ± 0.2 | 1.6 ± 0.1 | 13.6 ± 0.9 |
| 19 | 6.9 ± 0.2 | 1.1 ± 0.1 | 11.9 ± 0.4 | 52.3 ± 1.6 | 1.0 ± 0.1 | 5.5 ± 0.3 | 3.5 ± 0.3 | 1.4 ± 0.1 | 15.8 ± 1.0 |
| 20 | 6.8 ± 0.2 | 1.2 ± 0.1 | 11.3 ± 0.4 | 47.9 ± 1.5 | 1.3 ± 0.1 | 7.0 ± 0.4 | 4.3 ± 0.3 | 1.7 ± 0.1 | 17.6 ± 1.0 |
| 21 | 4.2 ± 0.1 | 4.2 ± 0.1 | 12.4 ± 0.5 | 53.4 ± 1.7 | 0.5 ± 0.1 | 0.9 ± 0.1 | 3.3 ± 0.2 | 1.8 ± 0.1 | 9.6 ± 0.6 |
| 22 | 2.0 ± 0.1 | 2.8 ± 0.1 | 9.2 ± 0.3 | 38.4 ± 1.3 | 0.5 ± 0.1 | 1.0 ± 0.1 | 2.4 ± 0.2 | 1.2 ± 0.1 | 10.7 ± 0.6 |
| 23 | 0.3 ± 0.1 | 1.8 ± 0.1 | 14.9 ± 0.5 | 62.8 ± 1.8 | 1.5 ± 0.2 | 10.5 ± 0.1 | 1.3 ± 0.2 | 1.9 ± 0.1 | 5.5 ± 0.4 |
| 24 | 0.3 ± 0.1 | 2.6 ± 0.1 | 13.1 ± 0.4 | 52.7 ± 1.6 | 1.6 ± 0.2 | 8.8 ± 0.4 | 1.4 ± 0.1 | 2.3 ± 0.2 | 7.3 ± 0.5 |
| 25 | 0.3 ± 0.1 | 0.5 ± 0.1 | 10.9 ± 0.3 | 44.3 ± 1.4 | 0.6 ± 0.1 | 8.8 ± 0.4 | 0.3 ± 0.1 | 1.1 ± 0.1 | 4.7 ± 0.3 |
| 26 | 0.3 ± 0.1 | 0.9 ± 0.1 | 9.6 ± 0.3 | 49.0 ± 1.5 | 1.1 ± 0.1 | 7.5 ± 0.4 | 0.6 ± 0.1 | 1.2 ± 0.1 | 19.8 ± 1.1 |
| 27 | 0.3 ± 0.1 | 0.6 ± 0.1 | 10.5 ± 0.3 | 74.7 ± 1.8 | 1.2 ± 0.1 | 6.7 ± 0.4 | 0.5 ± 0.1 | 1.2 ± 0.1 | 5.2 ± 0.4 |
| 28 | 0.6 ± 0.1 | 3.5 ± 0.1 | 15.1 ± 0.6 | 58.0 ± 1.6 | 0.9 ± 0.1 | 10.1 ± 0.5 | 0.9 ± 0.1 | 1.4 ± 0.1 | 9.5 ± 0.6 |
| 29 | 2.7 ± 0.1 | 2.7 ± 0.1 | 15.1 ± 0.6 | 55.1 ± 1.6 | 1.3 ± 0.1 | 6.4 ± 0.4 | 4.2 ± 0.3 | 1.4 ± 0.1 | 11.1 ± 0.6 |
| 30 | 5.4 ± 0.2 | 3.4 ± 0.1 | 16.3 ± 0.6 | 55.1 ± 1.6 | 0.4 ± 0.1 | 1.3 ± 0.1 | 1.9 ± 0.2 | 1.6 ± 0.1 | 11.5 ± 0.6 |

The rocks also contained rare Earth elements (REEs), as shown in Table 3. The identified REEs include $Rb_2O$, $YPO_4$, $ZrO_2$, $BaO$, $La_2O_3$, $Ce_2O_3$, $Nd_2O_3$, and $Sm_2O_3$. The most dominant of these REEs were $YPO_4$, $ZrO_2$, $La_2O_3$, and $Ce_2O_3$ with highest concentrations of 1235 $\mu g \cdot g^{-1}$, 2892 $\mu g \cdot g^{-1}$, and 23,570 $\mu g \cdot g^{-1}$, respectively.

The radioactivity measurement was carried out in rocks using an HPGe detector (Table 4). The results show that the Mamuju rocks are unique because they have very high concentrations of $^{238}U$ and $^{232}Th$ radionuclides. There are two categories (uranium prone and thorium prone) of rocks that have higher $^{232}Th$ concentrations, such as samples from Hulu Mamuju, Takandeang, and Rante Dunia, with a maximum $^{232}Th$ activity concentration of 288,639 $Bq \cdot kg^{-1}$. The rocks with the highest $^{238}U$ activity concentrations, such as Botteng, Hulu Ampalas, Salumati, and Ahu, were found in the Botteng region with a maximum activity concentration of 128,699 $Bq \cdot kg^{-1}$. The highest level of $^{40}K$ was measured in the Hulu Mamuju area and was approximately 23,088 $Bq \cdot kg^{-1}$.

**Table 3.** Rare Earth elements in rock samples.

| Samples | $Rb_2O$ $\mu g \cdot g^{-1}$ | $YPO_4$ $\mu g \cdot g^{-1}$ | $ZrO_2$ $\mu g \cdot g^{-1}$ | $La_2O_3$ $\mu g \cdot g^{-1}$ | $Ce_2O_3$ $\mu g \cdot g^{-1}$ | $Nd_2O_3$ $\mu g \cdot g^{-1}$ | $Sm_2O_3$ $\mu g \cdot g^{-1}$ |
|---|---|---|---|---|---|---|---|
| 1 | 175 ± 11 | 1094 ± 103 | 2232 ± 213 | 5025 ± 501 | 11,010 ± 1090 | 2921 ± 290 | 347 ± 32 |
| 2 | 504 ± 27 | 1235 ± 119 | 2892 ± 276 | 630 ± 61 | 2284 ± 225 | 524 ± 51 | 34 ± 3 |
| 3 | 301 ± 15 | 725 ± 69 | 1261 ± 125 | 9959 ± 989 | 23,570 ± 2257 | 5734 ± 571 | 647 ± 59 |
| 4 | 340 ± 17 | 795 ± 75 | 1535 ± 149 | 7232 ± 719 | 17,410 ± 1723 | 4466 ± 445 | 486 ± 47 |
| 5 | 337 ± 17 | 925 ± 90 | 1580 ± 155 | 7878 ± 778 | 17,450 ± 1723 | 4445 ± 432 | 555 ± 52 |
| 6 | 307 ± 15 | 827 ± 80 | 1825 ± 181 | 391 ± 37 | 728 ± 71 | *BLD | 73 ± 6 |

**Table 3.** *Cont.*

| Samples | Rb₂O μg·g⁻¹ | YPO₄ μg·g⁻¹ | ZrO₂ μg·g⁻¹ | La₂O₃ μg·g⁻¹ | Ce₂O₃ μg·g⁻¹ | Nd₂O₃ μg·g⁻¹ | Sm₂O₃ μg·g⁻¹ |
|---|---|---|---|---|---|---|---|
| 7 | 22 ± 2 | 80 ± 7 | 126 ± 12 | 701 ± 69 | 1433 ± 142 | 206 ± 19 | 102 ± 9 |
| 8 | 74 ± 5 | 89 ± 7 | 246 ± 23 | 438 ± 42 | 792 ± 78 | 137 ± 12 | 81 ± 7 |
| 9 | 48 ± 3 | *BLD | 126 ± 12 | 352 ± 33 | 639 ± 62 | 122 ± 11 | 59 ± 4 |
| 10 | 65 ± 5 | *BLD | 136 ± 13 | 442 ± 42 | 773 ± 76 | 167 ± 15 | 79 ± 6 |
| 11 | 381 ± 17 | *BLD | 120 ± 11 | 3174 ± 315 | 1610 ± 160 | 1172 ± 115 | 310 ± 29 |
| 12 | 397 ± 18 | 636 ± 60 | 1076 ± 104 | 3238 ± 321 | 1260 ± 125 | 1171 ± 115 | 293 ± 27 |
| 13 | 342 ± 16 | 89 ± 8 | 525 ± 51 | 3036 ± 301 | 1206 ± 120 | 1069 ± 101 | 293 ± 27 |
| 14 | 415 ± 19 | 21 ± 2 | 345 ± 32 | 3153 ± 314 | 1179 ± 116 | 1083 ± 103 | 290 ± 27 |
| 15 | 54 ± 3 | 90 ± 8 | 668 ± 63 | 393 ± 38 | 694 ± 68 | 156 ± 14 | 75 ± 6 |
| 16 | 2563 ± 235 | *BLD | 126 ± 12 | 808 ± 79 | 1497 ± 146 | 277 ± 25 | 32 ± 3 |
| 17 | 199 ± 15 | 208 ± 19 | 120 ± 11 | 453 ± 44 | 782 ± 76 | 84 ± 8 | 64 ± 6 |
| 18 | 953 ± 92 | 169 ± 15 | 56 ± 5 | 349 ± 33 | 1059 ± 104 | 44 ± 3 | 80 ± 8 |
| 19 | 1611 ± 149 | 67 ± 6 | 365 ± 34 | 521 ± 51 | 859 ± 83 | 74 ± 6 | 55 ± 5 |
| 20 | 674 ± 61 | 81 ± 8 | 366 ± 34 | 659 ± 64 | 1069 ± 105 | 54 ± 5 | 84 ± 8 |
| 21 | 191 ± 17 | 245 ± 21 | 128 ± 12 | 396 ± 37 | 759 ± 74 | 185 ± 17 | 79 ± 7 |
| 22 | 779 ± 71 | 226 ± 21 | 560 ± 55 | 690 ± 69 | 1414 ± 138 | 328 ± 31 | 46 ± 4 |
| 23 | 462 ± 45 | 820 ± 81 | 750 ± 73 | 326 ± 31 | 671 ± 66 | 175 ± 16 | 71 ± 6 |
| 24 | 651 ± 61 | 413 ± 40 | 756 ± 74 | 343 ± 33 | 737 ± 72 | 199 ± 18 | 85 ± 8 |
| 25 | 448 ± 42 | 725 ± 71 | 1261 ± 125 | 159 ± 15 | 313 ± 30 | 60 ± 5 | 41 ± 4 |
| 26 | 303 ± 29 | 621 ± 60 | 1001 ± 98 | 206 ± 19 | 436 ± 42 | 79 ± 7 | 80 ± 8 |
| 27 | 244 ± 21 | 802 ± 79 | 1123 ± 111 | 182 ± 17 | 372 ± 35 | 116 ± 10 | 34 ± 3 |
| 28 | 501 ± 49 | 645 ± 59 | 803 ± 80 | 322 ± 31 | 637 ± 61 | 150 ± 14 | 62 ± 6 |
| 29 | 822 ± 80 | 751 ± 72 | 1433 ± 141 | 215 ± 20 | 396 ± 38 | 63 ± 5 | 52 ± 5 |
| 30 | 969 ± 93 | 421 ± 40 | 862 ± 85 | 361 ± 35 | 541 ± 52 | 27 ± 2 | *BLD |

*BLD = below limit of detection.

**Table 4.** Radionuclide elements in rock samples.

| Samples | Gamma Spectrometry | | | | XRF | |
| | ²³⁸U Bq·kg⁻¹ | ²³²Th Bq·kg⁻¹ | ⁴⁰K Bq·kg⁻¹ | ²³⁵U Bq·kg⁻¹ | Total U μg·g⁻¹ | Total Th μg·g⁻¹ |
|---|---|---|---|---|---|---|
| 1 | 79,333 ± 6347 | 36,756 ± 2940 | 3078 ± 246 | 3598 ± 255 | 5363 ± 590 | 7633 ± 840 |
| 2 | 110,767 ± 8861 | 288,639 ± 23,091 | 23,088 ± 1847 | 4564 ± 412 | 5209 ± 573 | 52,730 ± 5300 |
| 3 | 15,667 ± 1253 | 211,271 ± 16,902 | 100 ± 8 | 660 ± 60 | 1444 ± 159 | 30,900 ± 3100 |
| 4 | 20,494 ± 1640 | 178,440 ± 14,275 | 100 ± 7 | 867 ± 81 | 1197 ± 132 | 29,210 ± 2903 |
| 5 | 50,113 ± 4000 | 261,266 ± 20,901 | 2182 ± 175 | 2165 ± 201 | 2084 ± 229 | 33,660 ± 3650 |
| 6 | 128,699 ± 10,296 | 1435 ± 115 | 77 ± 6 | 5663 ± 526 | 5450 ± 550 | 513 ± 51 |
| 7 | 2210 ± 177 | 1703 ± 136 | 669 ± 54 | 100 ± 8 | 505 ± 56 | 663 ± 66 |
| 8 | 565 ± 45 | 1374 ± 110 | 419 ± 34 | 25 ± 1 | 593 ± 58 | 644 ± 63 |
| 9 | 2230 ± 178 | 1919 ± 154 | 495 ± 40 | 111 ± 8 | 216 ± 21 | 506 ± 49 |
| 10 | 864 ± 69 | 984 ± 79 | 1280 ± 102 | 39 ± 2 | 194 ± 18 | 611 ± 60 |
| 11 | 1009 ± 81 | 2736 ± 219 | 633 ± 51 | 46 ± 3 | 153 ± 15 | 861 ± 85 |
| 12 | 990 ± 79 | 2619 ± 209 | 645 ± 52 | 47 ± 3 | 146 ± 14 | 864 ± 86 |
| 13 | 855 ± 68 | 2751 ± 220 | 639 ± 51 | 38 ± 3 | 108 ± 10 | 658 ± 63 |
| 14 | 815 ± 65 | 2594 ± 208 | 636 ± 52 | 35 ± 2 | 113 ± 11 | 713 ± 70 |
| 15 | 426 ± 34 | 959 ± 77 | 69 ± 6 | 16 ± 1 | 212 ± 20 | 611 ± 60 |
| 16 | 864 ± 69 | 576 ± 46 | 2290 ± 183 | 43 ± 3 | 117 ± 11 | 349 ± 32 |
| 17 | 373 ± 30 | 537 ± 43 | 674 ± 54 | 18 ± 1 | 64 ± 6 | 246 ± 23 |
| 18 | 539 ± 43 | 1005 ± 80 | 1540 ± 123 | 24 ± 2 | 73 ± 7 | 694 ± 67 |
| 19 | 864 ± 69 | 984 ± 79 | 1280 ± 102 | 35 ± 2 | 107 ± 10 | 314 ± 31 |
| 20 | 768 ± 61 | 1212 ± 95 | 1490 ± 119 | 35 ± 2 | 105 ± 10 | 1032 ± 99 |
| 21 | 14,738 ± 1179 | 826 ± 66 | 159 ± 13 | 621 ± 52 | 1446 ± 140 | 434 ± 41 |
| 22 | 5553 ± 444 | 1052 ± 84 | 318 ± 25 | 241 ± 19 | 1001 ± 99 | 394 ± 37 |
| 23 | 10,659 ± 853 | 553 ± 44 | 2594 ± 208 | 421 ± 38 | 801 ± 80 | 195 ± 18 |
| 24 | 22,538 ± 1804 | 623 ± 50 | 2320 ± 186 | 992 ± 71 | 2012 ± 199 | 229 ± 20 |
| 25 | 37,553 ± 3004 | 521 ± 42 | 3098 ± 248 | 1625 ± 141 | 2465 ± 23 | 143 ± 13 |
| 26 | 19,293 ± 1543 | 430 ± 34 | 1658 ± 133 | 821 ± 69 | 1209 ± 120 | 116 ± 10 |
| 27 | 92,009 ± 7361 | 471 ± 38 | 1668 ± 133 | 3956 ± 355 | 5306 ± 529 | 134 ± 12 |
| 28 | 9109 ± 729 | 585 ± 47 | 2296 ± 184 | 373 ± 31 | 925 ± 91 | 249 ± 23 |
| 29 | 27,420 ± 2194 | 954 ± 76 | 1491 ± 119 | 1115 ± 99 | 2496 ± 242 | 228 ± 20 |
| 30 | 29,154 ± 2332 | 680 ± 54 | 280 ± 22 | 1341 ± 121 | 2597 ± 251 | 405 ± 40 |
| Ave. | 22,882 ± 1602 | 33,549 ± 2348 | 1909 ± 134 | 988 ± 78 | 1457 ± 117 | 5531 ± 443 |
| WA | 37 ± 4 | 33 ± 3 | 400 ± 24 | - | - | - |

XRF = X-ray fluorescence; Ave. = average; WA = worldwide average in the soil [21].

## 4. Discussion

The diverse rock types in a volcanic complex indicate the existence of various factors that influence the formation of these rocks, including the source of magma, crystallisation rate, magma differentiation system, time, temperature, and pressure of magma formation, the amount of influential volatiles, and the tectonic arrangement of volcanic arcs. The results of elemental content measurements in the Mamuju sample using XRF can be seen in Table 2. It is known that rocks in Mamuju are dominated by elements of $Fe_2O_3$, $SiO_2$, $Al_2O_3$, and $Na_2O$, with several other major elements such as $MnO$, $TiO_2$, and $CaO$. Mamuju rocks also contain rare Earth elements or rare Earth metals (Table 3). The basaltic rocks that form the Adang volcanic rock complex are ultrapotassic rocks. Ultrapotassic rocks contain high levels of $K_2O$ when associated with shoshonitic and calc-alkaline series. According to the results of correlation analysis between $^{238}U$, $^{232}Th$ and the major elements in these rocks, there were several relationships of $Fe_2O_3$, $SiO_2$, $Al_2O_3$, and $Na_2O$ with U and Th in Mamuju (Figure 3). The correlation accumulation of U and Th levels also occurs in volcanic rocks that have undergone weathering and enrichment processes. This is in accordance with previous studies showing that the Mamuju area contains the radioactive materials thorianite (12% Pb), davidite (15% Fe), gummite (>70% Th), and autunite [4,5,7].

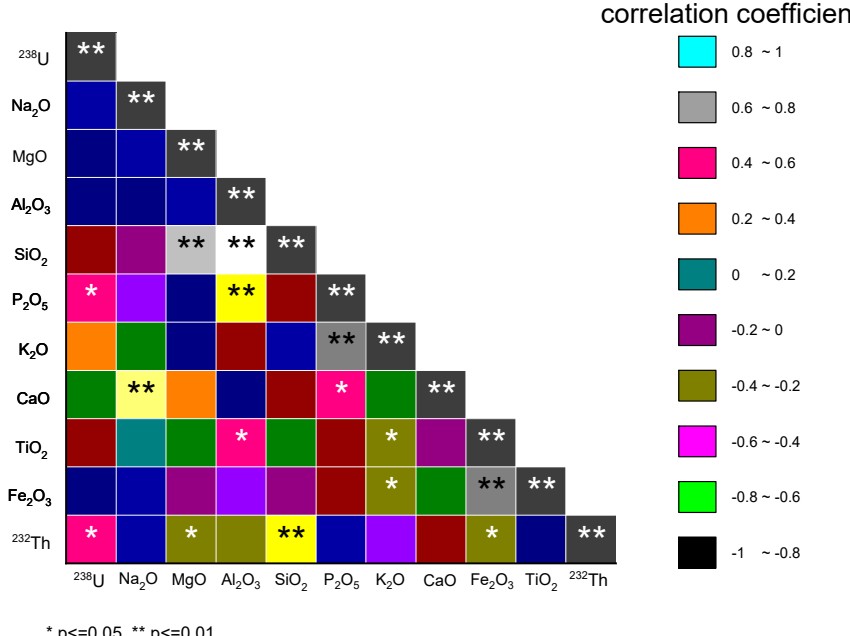

**Figure 3.** Correlation analysis between $^{232}U$ and $^{232}Th$ and major elements.

Among the REEs, there is a positive correlation between $YPO_4$ and $ZrO_2$ and between $La_2O_3$ and $Ce_2O_3$ (Figure 4). This is because $YPO_4$ ($^{91}Y$) is the parent radionuclide and emits beta particles into $ZrO_2$ ($^{91}Zr$) as in the decay scheme. Similarly, $La_2O_3$ ($^{140}La$) decays to $Ce_2O_3$ ($^{140}Ce$). With such conditions, Mamuju rocks contain higher levels of zircon and cerium.

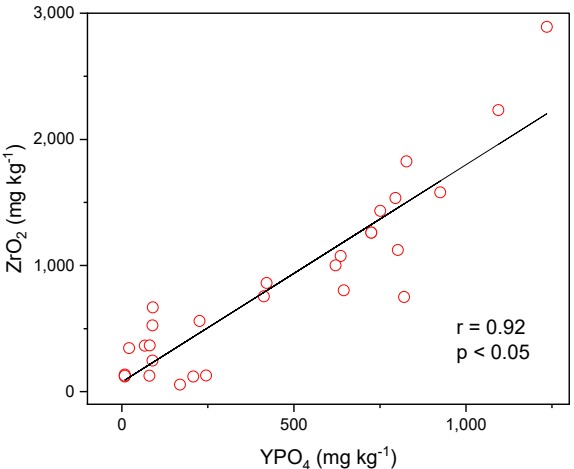

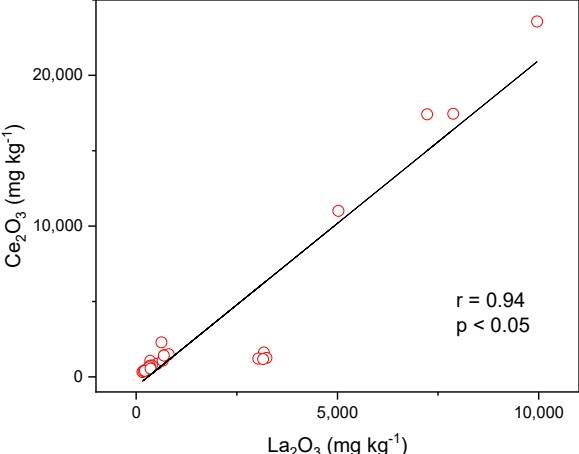

**Figure 4.** Correlation between $ZrO_2$ and $YPO_4$ concentrations and between $Ce_2O_3$ and $La_2O_3$ concentrations.

The results of HPGe analysis show that Mamuju rocks have a very high concentration of [238]U and [232]Th radionuclides. On the basis of the concept of secular equilibrium, concentrations of [238]U and [226]Ra are considered the same concentration between the parent and the progeny. We created a comparison chart of [238]U activity concentrations using an HPGe detector and for total U using XRF analysis (Figure 5). This method was also used to compare [232]Th and total Th. Correlation analysis was conducted using Pearson correlation. The *p*-value was less than 0.05 (two-tailed) and, thus, it was categorised as having a significant correlation. The correlation coefficient values of the U and Th variables were 0.93 and 0.92, respectively, which shows that both methods have a strong positive correlation. This graph shows that there is a uniformity between the [238]U results of HPGe analysed from the energy peak [214]Pb and [214]Bi and the concentration of total U measured by XRF and between the [232]Th results of HPGe analysed from the energy peak [212]Pb, [208]Tl, and [228]Ac and the concentration of total Th measured by XRF.

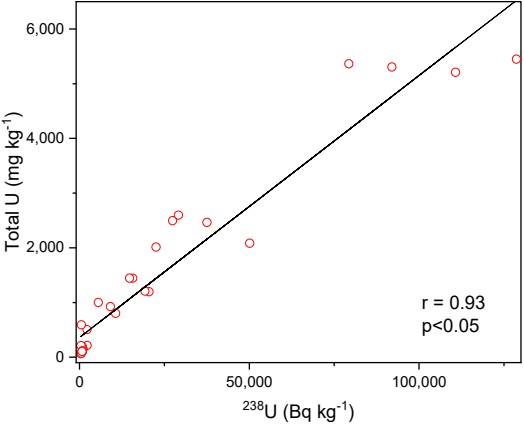

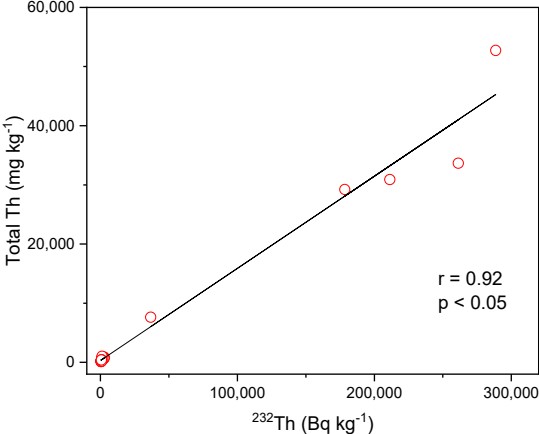

**Figure 5.** Comparison of $^{238}$U and $^{232}$Th concentration between gamma spectrometry and XRF measurement.

According to the HPGe measurement result, Mamuju is an anomalous area because it has high contents of $^{238}$U, $^{232}$Th, and $^{40}$K with activity concentrations of 539–128,699 Bq·kg$^{-1}$ (average: 22,882 ± 1602 Bq·kg$^{-1}$), 471–288,639 Bq·kg$^{-1}$ (average: 33,549 ± 2348 Bq kg$^{-1}$), 69–23,088 Bq·kg$^{-1}$ (average: 1909 ± 134 Bq·kg$^{-1}$), respectively. According to the United Nations Scientific Committee on the Effects of Atomic Radiation (UNSCEAR) data, these values exceed the global elemental average concentrations of soil samples; the average concentration of $^{238}$U is around 37 ± 4 Bq·kg$^{-1}$, that of $^{232}$Th is around 33 ± 3 Bq·kg$^{-1}$, and that of $^{40}$K is around 400 ± 24 Bq·kg$^{-1}$ (Figure 6) [21]. This shows that Mamuju has potential as an exploration area for radioactive minerals if we compare these values with other studies [22–24].

According to the measurement results, rock samples from Mamuju have a $^{238}$U/$^{235}$U AR of 20.1–26.6, with an average of 22.9. This value indicates that the Mamuju area is still in natural condition without the effect of fallout from other sources (Figure 7) [11,15].

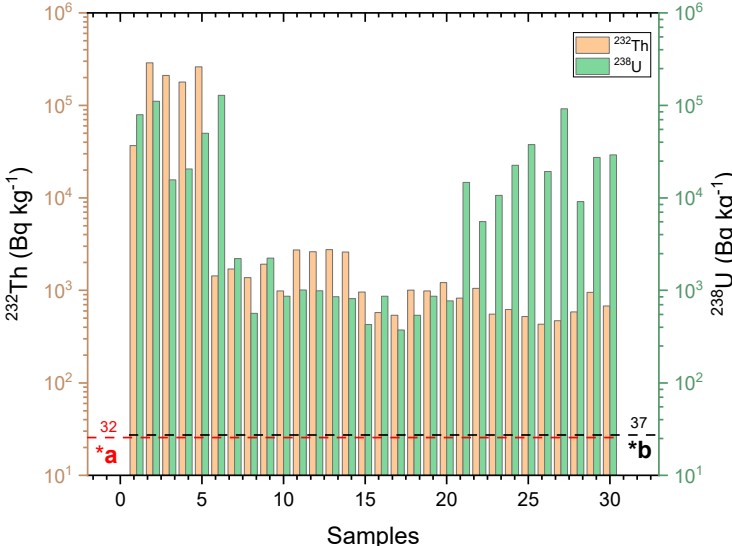

**Figure 6.** $^{238}$U and $^{232}$Th concentration in Mamuju. *a is the worldwide average for $^{238}$U; *b is the worldwide average for $^{232}$Th.

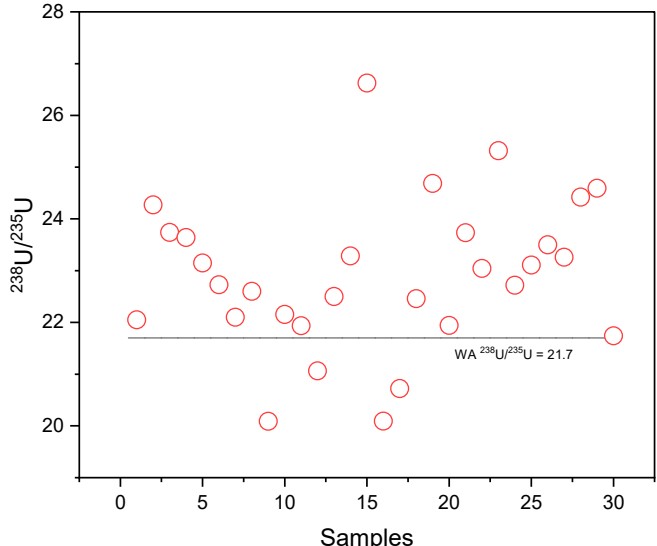

**Figure 7.** $^{238}$U/$^{235}$U activity ratio. WA is the natural world average of $^{238}$U/$^{235}$U activity ratio.

## 5. Conclusions

Mamuju is an anomalous area due to its high content of U and Th concentrations. From this study, it can be concluded that Mamuju rocks have $^{238}$U concentration characteristics of 539–128,699 Bq·kg$^{-1}$ and $^{232}$Th concentration characteristics of 471–288,639 Bq·kg$^{-1}$. The U occurring in the region is categorised as natural with an average AR $^{238}$U/$^{235}$U of 22.9. The major elements include $Fe_2O_3$, $SiO_2$, $Al_2O_3$, and $Na_2O$, with several other major elements such as $MnO$, $TiO_2$, and $CaO$ and REE elements such as $ZrO_2$ and $Ce_2O_3$. Therefore, Mamuju has potential as an exploration area for radioactive minerals.

**Author Contributions:** Conceptualization, E.D.N., M.H., and S.T.; methodology, E.D.N. and N.A.; validation, I.R. and E.D.N.; formal analysis, E.D.N. and I.R.; investigation, H.S., I.R., and E.D.N.; resources, E.D.N., I.R., H.S., and S.T.; data curation, E.D.N. and I.R.; writing—original draft preparation, E.D.N. and I.R.; writing—review and editing, E.D.N., I.R., H.S., S.P., M.H., N.A., and S.T.; visualization, E.D.N.; supervision, H.S., S.P., M.H., N.A., and S.T.; funding acquisition, H.S. and S.T. All authors read and agreed to the published version of the manuscript.

**Funding:** This research was partially funded by the Indonesian government (DIPA 2014-2019).

**Conflicts of Interest:** The authors declare no conflict of interest.

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
