# Peer review of "Natural Radioactivity of Laterite and Volcanic Rock Sample for Radioactive Mineral Exploration in Mamuju, Indonesia"

_geosciences, doi:10.3390/geosciences10090376_

Round 1

Reviewer 1 Report

Rosianna et al.

While the paper is a good short description of the samples, it is very different from most geological work on uranium deposits. the XRF section is fine but in my opinion the gamma section is flawed.

In most work the object of the gamma ray study is to determine equivalent uranium (and similarly Th and K) by measuring using a gamma channel in a spectrometer calibrated using a sample(s) that is in equilibrium. Thus results are reported as eppm U based on 214 Bi channel or eTh for 208 Tl rather than Bq kg-1. This would much improve this paper. Reports of 238U from gamma ray studies are incorrect as this assumes equilibrium.

Most uranium exploration involves airborne gamma-ray measurements and ground follow-up. This study appears a very limited coverage of this.

Author Response

Dear Editor and Reviewers,

We thank the editor and reviewers for careful reading of our manuscript and for giving useful comments. Our responses are given below.

 [Reviewer #1]

While the paper is a good short description of the samples, it is very different from most geological work on uranium deposits.

The XRF section is fine, but in my opinion, the gamma section is flawed.  In most work, the object of the gamma-ray study is to determine equivalent uranium (and similarly Th and K) by measuring using a gamma channel in a spectrometer calibrated using a sample(s) that is in equilibrium. Thus results are reported as eppm U based on 214Bi channel or eTh for 214Tl rather than Bq kg-1. This would much improve this paper. Reports of 238U from gamma-ray studies are incorrect as this assumes equilibrium. Most uranium exploration involves airborne gamma-ray measurements and ground follow-up. This study appears a very limited coverage of this.

Ans. Thank you very much for your constructive comments. This study uses a gamma spectrometer as a non-destructive and fast technique for NORM samples. However, we are use gamma spectrometry using the HPGe detector, which has better capabilities than the scintillation detector (NaI(Tl)). The HPGE detector has an excellent resolution around 2 keV so that we can analyse a sample with multi peaks from 238U or 232Th progeny. Such as from 214Pb (351 keV), 214Bi (609 keV), 234mPm (1001 keV) and 234Th (63 or 93 keV) were identified for calculations of 238U. Likewise, for Thorium, we use 212Pb (238 keV), 208Tl (581 keV) and 228Ac (911 keV). We believe that the low-resolution scintillation detector (NaI(Tl)) lacks this capability.

As you know, radioactivity measurements of NORM samples using a gamma spectrometer (HPGe or NaI(Tl)) generally used the principle of equilibrium between the progeny and the parent radionuclides. This is because the parent radionuclides have a long half-life, so we use the progeny peak(s) in the decay row and assuming the equilibrium condition (secular equilibrium) has occurred. In this study, we analysed rock samples and analysed them in the laboratory (not in situ measurements) using HPGe detector, then the samples were prepared, which were placed in the vial and closed tightly. The tight closure of this vial aims to create equilibrium between the parent and the progeny, of which one of the decays of U and Th is 222Rn and 220Rn which are gaseous. So, we believe that our measurement technique is more accurate. Where from much literature/references for NORM samples, there will be secular equilibrium about 30 days. In rock sample, generally, 238U and 226Ra are equilibrium. Base on that assumption, we have choice peaks of 214Bi and 214Pb to determination of 238U. Another option to determination of 238U such as 234mPm or 234Th. However, “they” will have secular equilibrium between parents and progeny around 90 days. With this response letter, here we are adding some references to our method regarding the equilibrium condition.

  1. International Atomic Energy Agency. IAEA-TECDOC-1363. Guidelines for radioelement mapping using gamma ray spectrometry data. IAEA; Vienna, Austria, 2003
  2. Hassan, N.M., Kim, Y.J., Jang, J. et al. Comparative study of precise measurements of natural radionuclides and radiation dose using in-situ and laboratory γ-ray spectroscopy techniques. Sci Rep 8, 14115 (2018). https://doi.org/10.1038/s41598-018-32220-9
  3. Masod abdulqader, Sanna, et al. Natural radioactivity of intrusive-metamorphic and sedimentary rocks of the Balkan Mountain range (Serbia, Stara Planina). Minerals. 2018. 8(1), pp 6.
  4. Srinivasa, E., Rangaswamy, D.R. & Sannappa, J. Assessment of radiological hazards and effective dose from natural radioactivity in rock samples of Hassan district, Karnataka, India. Environ Earth Sci 78, 431 (2019). https://doi.org/10.1007/s12665-019-8465-z
  5. Murray A.S., Helsted L.M., Autzen M., Jain M., Buylaert J.P. Measurement of natural radioactivity: Calibration and performance of a high-resolution gamma spectrometry facility(2018) Radiation Measurements,  120 , pp. 215-220
  6. Xinwei Lu, Xiaolan Zhang, Natural radioactivity measurements in rock samples of Cuihua Mountain National Geological Park, China, Radiation Protection Dosimetry, Volume 128, Issue 1, January 2008, Pages 77–82, https://doi.org/10.1093/rpd/ncm236
  7. Nanping WANG , Lei XIAO , Canping LI , Ying HUANG , Shaoying PEI ,Shaomin LIU , Fan XIE & Yexun CHENG (2005) Determination of Radioactivity Level of 238U, 232Th and 40K in Surface Medium in Zhuhai City by in-situ Gamma-ray Spectrometry, Journal of Nuclear Science and Technology, 42:10, 888-896, DOI: 10.1080/18811248.2005.9711040

Meanwhile, the unit of radioactivity in general, if using fixed HPGe (not in situ) is Bq/kg (activity concentration). This is possible if we convert it to ppm (mass concentration) using several other parameters such as atomic weight and others.

1 Bq 238U kg-1 = 81 ppb U (81 10-9 gU g-1)

1 Bq 232Th kg-1  = 246 ppb Th (246 10-9 gTh g-1)

1 Bq 40K kg-1  = 32.3 ppm K (32.3 10-6 gK g-1)

 With this explanation, We have kept these manuscripts in this version.

Reviewer 2 Report

Maybe it could be useful a comparison between the level of radioactivity founded in your samples and some other extraction places in the world. At least for future works.

Author Response

Dear Editor and Reviewers,

We thank the editor and reviewers for careful reading of our manuscript and for giving useful comments. Our responses are given below.

[Reviewer #2]

Maybe it could be useful a comparison between the level of radioactivity founded in your samples and some other extraction places in the world. At least for future works.

Ans. First of all, thank you very much for reading our manuscript and giving us the comments for improvements. We will be considered in our future works.

Reviewer 3 Report

In this study, the radionuclides and other important elements were measured, to determine the characteristics of radionuclides and other mineral measurements, which are essential for categorising Mamuju rocks and exploring the region as a potential mining area.

This manuscript is a well-written study with a logical structure. The introduction provides sufficient background and includes all relevant references. Connecting to the "Material and Methods" and “Results” parts I found some small mistakes. The "Discussion" is detailed: I found some missing data and mistakes. English correction is necessary.

Comments and questions:

1. Introduction

a) What was the source of Figure 1.? I have checked the Reference-3 and I did not find inside.

2. Materials and methods:

a) Line 160: U-235 is missing

b) Statistical analysis: it is not too detailed, please develop this part.

c) You wrote that you collected “fresh” rock sample. What does it mean “fresh”? Why was it important?

3. Results:

a) Line 188, 192, 202 : Tabel 2,3,4 = Table 2,3,4. Please correct them

b) In Table 4 the worldwide average is for the activity concentration in the soil, right?

4. Discussion:

a) Figure 3: what is the difference between the colour of 0.8-1 and colour of 0.6-0.8 or between the colour of -0.8- -0.6 and colour of -1- -0.8? Because it seems they are the same. I recommend to change the colours on this Figure for that it will be clear. My other question for this Figure is what was the p-value of the cases without stars?

b) Line 256-257: What is about the Th? You wrote about only the U.

c) Line 272: 40Kactivity=40K activity. Please correct it.

d) What were the results of the U-235 measurements? In the Material and Methods you wrote, that it was determined, but the results are missing.

e) Line 291-297: it is not a conclusion, but a theoretical or experimental part. Please relocate it.

f) Line 299: What does it mean: +/- **?

g) What is your conclusion: natural AR is good or not?

Author Response

Dear Editor and Reviewers,

We thank the editor and reviewers for careful reading of our manuscript and for giving useful comments. Our responses are given below.

[Reviewer #3]

In this study, the radionuclides and other important elements were measured, to determine the characteristics of radionuclides and other mineral measurements, which are essential for categorising Mamuju rocks and exploring the region as a potential mining area. This manuscript is a well-written study with a logical structure. The introduction provides sufficient background and includes all relevant references. Connecting to the “Material and Methods” and “Results” parts, I found some small mistakes. The “Discussion” is detailed: I found some missing data and mistakes. English correction is necessary.

Ans. Thank you very much for your constructive comments on our manuscript. We have read your comments carefully and response one by one. Certificate of English correction is attached.

  1. Introduction
  2. a) What was the source of Figure 1.? I have checked the Reference-3, and I did not find inside.

Ans. Thank you very much for pointing out the mistake. We replaced as Reference-4.

  1. Materials and methods
  2. a) Line 160: U-235 is missing

Ans. Thank you very much for pointing out the mistake. We are adding the 235U on the sentence. Moreover, the sentence will be “The minimum detectable concentrations (MDCs) of 238U, 232Th, 235U, and 40K……….

  1. b) Statistical analysis: it is not too detailed, please develop this part.

Ans. Thank you very much for your constructive comments on our manuscript. We are adding the sentences “We conducted a Pearson correlation analysis and calculated the values’ Pearson correlation coefficients with two-tailed of significance test (p-value) at 0.05 and 0.01. The Equation (3) in expression for the Pearson correlation coefficients.

where n is a sample size, xi and yi are individual sample points indexed, and  are the sample mean”.

  1. c) You wrote that you collected “fresh” rock sample. What does it mean “fresh”? Why was it important?

Ans. Thank you very much for your constructive comments on our manuscript. Fresh rock the geologist’s term for rocks that do not appear weathered. The purpose takes the fresh rock to avoid weathering factors so that the fresh rock obtained is more following the actual rock conditions. We also added the sentence “The purpose of taking fresh rock is to avoid weathering factors so that the fresh rock obtained in accordance with the actual rock conditions”.

  1. Results:
  2. a) Line 188, 192, 202 : Tabel 2,3,4 = Table 2,3,4. Please correct them

Ans. Thank you very much for pointing out the mistake. We corrected the Tabel to Table.

  1. b) In Table 4, the worldwide average is for the activity concentration in the soil, right?

Ans. Thank you very much for your constructive comments on our manuscript. Yes, in the UNSCEAR report it shows the value for soil. So I added, “Word average of the soil” to avoid misunderstanding.

  1. Discussion:
  2. a) Figure 3: what is the difference between the colour of 0.8-1 and colour of 0.6-0.8 or between the colour of -0.8- -0.6 and colour of -1- -0.8? Because it seems they are the same. I recommend to change the colours on this Figure for that it will be clear. My other question for this Figure is what the p-value of the cases was without stars?

Ans. Thank you very much for your constructive suggestion on our manuscript. I already changed the Figure for make it easy to distinguish between ranges. The starless column shows that it has a p-value of more than 0.05 (> 0.05), which means that the value is not statistically significant and provides strong evidence for the null hypothesis.

  1. b) Line 256-257: What is about the Th? You wrote about only the U.

Ans. Thank you very much for your constructive comments on our manuscript. We added the sentences about the 232Th, and the sentence will be “This graph shows that there is a uniformity between the 238U results of HPGe analysed from the energy peak 214Pb and 214Bi with the concentration of total U measured by XRF and between the 232Th results of HPGe analysed from the energy peak 212Pb, 208Tl and 228Ac with the concentration of total Th measured by XRF.

  1. c) Line 272: 40Kactivity=40K activity. Please correct it.

Ans. Thank you very much for pointing out the mistake. I already fix them to be “…….238U, the 232Th and 40K activity concentration…….”

  1. d) What were the results of the U-235 measurements? In the Material and Methods you wrote, that it was determined, but the results are missing.

Ans. Thank you very much for your constructive comments on our manuscript. The results of U-235 was shown in table 4 (line 202), and we used it in the discussion section for determining AR.

  1. e) Line 291-297: it is not a conclusion, but a theoretical or experimental part. Please relocate it.

Ans. Thank you very much for your constructive comments on our manuscript. As reviewer #3 suggestion, the sentences moved to the introduction section (line 86-93) and we are updated the references list order.

  1. f) Line 299: What does it mean: +/- **?

Ans. Thank you very much for pointing out the mistake. Accordingly the reference its not have the standard deviation cause that We are deleted the +/- **.

  1. g) What is your conclusion: natural AR is good or not?

Ans. Thank you very much for your constructive comments on our manuscript. As we mention in the manuscript. To make it more clear we added the sentence in manuscript “This value are indicated that Mamuju area are still in natural condition without the effect of fallout from other sources (Figure. 7) [19, 23].

Round 2

Reviewer 1 Report

as previous review

some issues addressed but don't address equilibrium problem

Author Response

[Reviewer #1]

some issues addressed but don't address equilibrium problem

Ans. Thank you very much for your constructive comments. I already adding some sentences to provide information about the secular equilibrium. The sentence will be

“Approximately 100 g of each sample was weighed and sealed in U-8 container. The air tight of this vial aims to create equilibrium between the parent and the progeny, of which one of the decays of U and Th is 222Rn and 220Rn which are gaseous. Furthermore, the samples are left for more than 30 days to allow decay products to reach secular equilibrium before further gamma spectrometer measurements”

Reviewer 2 Report

When I said that the interest to the readers is average I thought about the narrow area of researchers that are looking for this subject ( in the actual context).

Author Response

[Reviewer #2]

When I said that the interest to the readers is average I thought about the narrow area of researchers that are looking for this subject ( in the actual context).

Ans. thank you very much for reading our manuscript.

Reviewer 3 Report

I have no more comments. 

Author Response

[Reviewer #3]

I have no more comments.

Ans. thank you very much for reading our manuscript.